# Genetic insights into number of long-term conditions and their relationship with lifespan

Youngjune Bhak[ORCID][1]*, Bruce Guthrie[2,3], Albert Tenesa[1,4]*

**1** MRC Human Genetics Unit at the MRC Institute of Genetics and Molecular Medicine, University of Edinburgh, Western General Hospital, Edinburgh, United Kingdom, **2** Usher Institute, University of Edinburgh, Edinburgh, United Kingdom, **3** Advanced Care Research Centre, University of Edinburgh, Edinburgh, United Kingdom, **4** The Roslin Institute, University of Edinburgh, Easter Bush Campus, Midlothian, United Kingdom

* ybhak@ed.ac.uk (YB); Albert.Tenesa@ed.ac.uk (AT)

## Abstract

### Aims

Relationships between the genetic risk for long-term conditions (LTCs) and lifespan have been reported. However, the genetic factors underlying the total number of LTCs an individual has (LTC burden) and their association with lifespan have not been fully investigated. This study aims to investigate the genetics of LTC burden and evaluate its relationship with lifespan.

### Methods

A genome-wide association study (GWAS) and a genetic heritability test were conducted on LTC burden using data from 343,868 UK Biobank individuals. Global and local genetic correlations between the LTC burden and parental lifespan were estimated. A polygenic risk score ($PRS_{LTC}$) for LTC burden was derived from a separate set of 34,339 UK Biobank individuals with records of age at death, who were not included in the GWAS analysis. The association between the $PRS_{LTC}$ and lifespan, as well as its ability to predict LTC burden, was assessed.

### Results

Loci in the *HLA* region were the most significant among the 21 significant independent loci from the GWAS. The estimated SNP heritability of LTC burden was 0.0963 and significantly different from zero (se = 0.0034, p-value = 1.77 x $10^{-176}$). The global genetic correlation between LTC burden and parental lifespan exhibited a significant global genetic correlation of −0.7869 (se = 0.0419, p-value 9.57 x $10^{-79}$). Additionally, 44 loci showed significant local genetic correlations (p-value < 2.23 x $10^{-5}$). Individuals in the highest 10% $PRS_{LTC}$ had, on average, a 0.9-year shorter lifespan and 0.73 more LTCs than those in the lowest 10%.

**Data availability statement:** The data used in this study are part of the UK Biobank resource. Details on procedures for accessing the UK Biobank data are available at https://www.ukbiobank.ac.uk/enable-your-research/apply-for-access. The genome-wide association study (GWAS) summary statistics generated and analysed in this study have been deposited in Zenodo under the DOI 10.5281/zenodo.17272288 and can be accessed without restriction for reuse and further analysis.

**Funding:** This project was funded by the National Institute for Health Research (NIHR) Artificial Intelligence and Multimorbidity: Clustering in Individuals, Space and Clinical Context (AIM-CISC) grant NIHR202639. The views expressed are those of the author(s) and not necessarily those of the NIHR or the Department of Health and Social Care. The funders had no role in study design, data collection and analysis, decision to publish, or preparation of the manuscript.

**Competing interests:** The authors have declared that no competing interests exist.

## Conclusions

This study identifies significant genetic factors associated with LTC burden and their association with lifespan, providing insights into the genetic underpinnings of both multiple LTCs and lifespan.

## Introduction

A number of long-term conditions (LTCs), which are chronic conditions requiring lifelong or long-term management, affect human life expectancy [1]. For example, heart failure is associated with a loss of 7.3–20.5 years of life expectancy [2], type 2 diabetes with a loss of 0.99–7.67 years [3], and serious mental illness with a loss of 15–20 years [4]. Multimorbidity, defined as the co-occurrence of multiple LTCs (MLTC), is also associated with a 4.54-5.15-year reduction in life expectancy [5]. The cumulative number of LTCs an individual has, referred to as LTC burden, provides a way to capture overall morbidity load and its potential influence on survival.

There is evidence that genetic factors contribute to both individual LTCs and lifespan. Previous studies have shown that polygenic risk scores (PRSs), which quantify genetic risks for complex diseases and traits, are associated with differences in life expectancy [6], and genome-wide analyses have identified loci associated with human longevity, including the Human Leukocyte Antigen (HLA) region and the Lipoprotein (a) (LPA) gene locus [7,8]. However, most research has focused on single diseases or broad measures of longevity, leaving the genetics of LTC burden and its relationship with lifespan underexplored.

Understanding the shared genetic architecture of LTC burden and lifespan may provide insights into the biological mechanisms that influence both multimorbidity and longevity. Immune dysregulation and chronic inflammation, captured by HLA variation, and lipid metabolism, influenced by LPA, are two plausible biological pathways implicated in these processes [9–11]. Investigating the extent to which LTC burden shares genetic factors with respect to lifespan can help identify potential pathway linking multimorbidity and aging.

In this study, we investigated the genetic basis of LTC burden and its association with lifespan using large-scale UK Biobank data. We performed a genome-wide association study (GWAS) of LTC burden, estimated SNP heritability, and examined both global and local genetic correlations with parental lifespan, which serves as a validated proxy variable for lifespan [12–15]. Finally, we derived a PRS for LTC burden ($PRS_{LTC}$) in 34,339 UK Biobank individuals, who were not included in the GWAS, and evaluated its association with both LTC burden and lifespan to assess the genetic underpinnings of LTC burden and their potential contribution to reduced longevity.

## Materials and methods

### Data source

The UK Biobank (UKB) is a prospective, population-based cohort study that includes comprehensive phenotype and genotype data from approximately 500,000

participants recruited between 2006 and 2010 residing in England, Scotland, and Wales (www.ukbiobank.ac.uk). This open-access resource was established to support investigations into the factors influencing various health outcomes [16]. To ensure homogeneity, we selected unrelated individuals of white ancestry (UK Biobank Data-Field 22006) with less than 10% missing genotypes and with concordance between recorded sex and genetically determined sex. The unrelated participants were identified and extracted using the KING software with the following options: --unrelated --degree2 (version 2.28) [17]. Genotypes of the selected individuals were filtered using PLINK software (version 1.90b) with the following options: --geno 0.01, --hwe 1e-15, --maf 0.01, and mind 0.1, retaining 525,262 variants [18]. Following these selections, the dataset was divided into two sets: 343,868 individuals without a recorded age at death, designated as the training set, and 34,339 individuals with a recorded age at death, designated as the validation set. The training set was used to identify genetic associations and construct polygenic scores for long-term conditions (LTCs), whereas the validation set was used to apply and validate these scores in relation to lifespan.

### Ethics statement

The UK Biobank project was approved by the National Research Ethics Service Committee North West-Haydock (REC reference: 11/NW/0382). Participants provided written informed consent to participate in the UK Biobank. An electronic signed consent was obtained from the participants. This research was conducted using the UK Biobank Resource under projects 44986.

### Study outcomes: LTC classification and characterization

The data including participants' characteristics and LTCs in the present study were accessible via the UK Biobank Study [16]. The LTC burden in the present study was derived by counting the number of LTCs that each participant had, recorded in hospital inpatient records (UK Biobank Data field 41270). The 32 LTCs examined in this study were defined and categorised based on ICD-10 codes as follows: active asthma (J45 and J46), alcohol problems (E244, F101-F109, G312, G621, G721, I426, K292, K70, K852, K860, Z502, and Z714), anorexia and bulimia (F500-F503), anxiety (F40 and F41), atrial fibrillation (I48), blindness (H540-H542), bronchiectasis (J47 and Q334), cancers (C00-C25, C260, C261, C268, C269, C30-C87, C880, C882, C90-C97, D05, D45, and D46), coronary heart disease (I200, I201, I208, I209, and I21-I25), chronic kidney disease (I120, I131, I132, N183, and N184), chronic obstructive pulmonary disease (J40-J44), chronic liver disease (B150, B160, B190, I85, I982, I983, K70, K700, K704, K711, K72, K743, K754, K758, K760, K762, K763, and K766), dementia (F00, F01, F03, F051, and G30), depression (F32 and F33), diabetes (E10-E14), diverticular disease (K382 and K57), epilepsy (G40 and G41), heart failure (I110, I130, I132, and I50), hypertension (I10-I15), Inflammatory bowel disease (K50, K51, M074, M075, and M076), learning disability (F70-F73, F78, F79, F819, and Q90), multiple sclerosis (G35), psychoactive misuse (F11-F19), Parkinson's disease (F023 and G20), peripheral vascular disease (I731, I738, I739, I743, I744, and I745), hyperplasia of prostate (N40), rheumatoid arthritis (L405, M02, M03, M070, M071, M072, M073, M08, M09, M315, M316, M353, and M45), schizophrenia and bipolar (F30 and F31), stroke and transient ischemic attack (G450-G454, G458, G459-G468, I61, I630-I635, I638, I639, I64-I66, I691, I693, and I694), thyroid disorders (E035, E038, E039, E050-E052, E055, E058, E059, E062, E063, E065, E069, and H062), dyspepsia (K227, K25-K28), and viral hepatitis (B18) [19].

### Genetic analysis

To investigate genetics of LTC burden, we conducted a GWAS using the training set. For the GWAS, REGENIE software (version 3.2.2) was utilized with default options, except for the following options applied in both step 1 and 2: --apply-rint --qt --maxCatLevels 40 [20]. The block sizes were set to --bsize 1000 for step 1 and --bsize 400 for step 2. Covariates considered in the GWAS include age, age square, sex, age*sex, age square*sex, UK Biobank Centre, UK Biobank genetic array, and the first 20 genetic principal components. To derive independent genetic loci, the GWAS result was

clumped to extract index variants using PLINK software with the following options: --clump, --clump-p1 0.00000005, --clump-r2 0.001, and --clump-kb 10000 [18].

To estimate the genetic heritability of LTC burden, we used LD-score regression (version 1.0.1) [21] with default options. LD scores were estimated from the quality-controlled genotype data described above using the following options: --ld-wind-cm 1.0 --l2. An LDSC regression intercept greater than 1.05 was considered indicative of potential residual population stratification, model misspecification, or other artefacts. In our analysis, the intercept was 0.89, indicating no evidence of inflation. The genetic heritability was derived using default options. To investigate the genetic correlation between LTC burden and lifespan, we used GWAS results on parental lifespan from large biobank studies, taking it as a surrogate variable for lifespan [12]. Global and local genetic correlations between were estimated using High-Definition Likelihood (HDL, version 1.4.0) and LAVA software (version 0.1.0), respectively with default option [22,23]. LD references to use in HDL were generated from the quality-controlled genotype data using build_ld_ref functions of HDL software with default options. For LAVA, we used LD reference data (g1000_eur) and locus definition file (blocks_s2500_m25_f1_w200. GRCh37_hg19.locfile covering 2,495 regions) provided by LAVA. Only 2,247 of the 2,496 regions were able to be analysed and a Bonferroni-corrected significance threshold of $p < 0.05/2,247$ ($2.23 \times 10^{-10}$) was applied to account for the number of tests accordingly. To investigate the genetic risk of LTC burden in relation to lifespan, we derived a PRS for LTC burden in the validation set by using the GWAS result on LTC burden in the training set using PRSice-2 software (version 2.3.3) with options specifying column names and --binary-target F [24].

## Statistical analyses

Categorical variables were presented as counts and percentages, and continuous variables were presented as means and standard deviations (SD). The correlations between the PRS and both LTC burden and lifespan were assessed using Spearman's rank-order correlation. LTC burden and lifespan between the PRS groups were compared using Wilcoxon signed-rank test. To account for the multiple tests, we applied thresholds for significance based on Bonferroni correction. All statistical analyses not performed using the software described were conducted in R (version 4.4.3).

## Results

A total of 343,868 individuals who were alive at the end of follow-up comprised the training set, and 34,339 individuals known to have died, with records of age at death, comprised the validation set (S1 Table). At recruitment to the UK Biobank study, the average age in the training set was 56.5 years (SD 7.9) compared with 61.9 years (SD 6.3) in the validation set. The training set had a lower proportion of males than the validation set (44.9% vs 59.7%). Similarly, the average number of LTCs was 1.41 (SD 1.77) in the training and 3.54 (SD 2.44) in the validation set. The average age at death in the validation set was 71.2 years (SD 7.5), with males averaging 71.0 years (SD = 7.63) and females 71.3 years (SD = 7.33). The difference between sexes was statistically significant (p-value = $5.0 \times 10^{-4}$).

To investigate the genetics of LTC burden, we conducted a GWAS using the training set. After LD-based clumping, 21 independent genetic loci were identified (Fig 1A, S2 Table). The most significant association, rs532965, was observed in the intron of human leukocyte antigen class II gene *HLA-DRB1* (S3 Table). The SNP heritability of LTC burden was modest but statistically significant, estimated at 0.0963 (se = 0.0034, p-value = $1.77 \times 10^{-176}$, regression intercept = 0.887 (0.101)).

To assess global and local genetic correlation between LTC burden and lifespan, we used a previously published GWAS result of parental lifespan as a surrogate for lifespan. The global genetic correlation between LTC burden and lifespan was −0.7869 (se = 0.0419, p-value $9.57 \times 10^{-79}$). There were 44 significant negative local genetic correlations between LTC burden and lifespan out of the 2,247 loci assessed (p-value < $2.23 \times 10^{-5}$, Fig 1B, S4 Table). The most significant negative local correlation encompassed the lipoprotein (a) (LPA) gene region (chr6: 160,583,919–161,371,014).

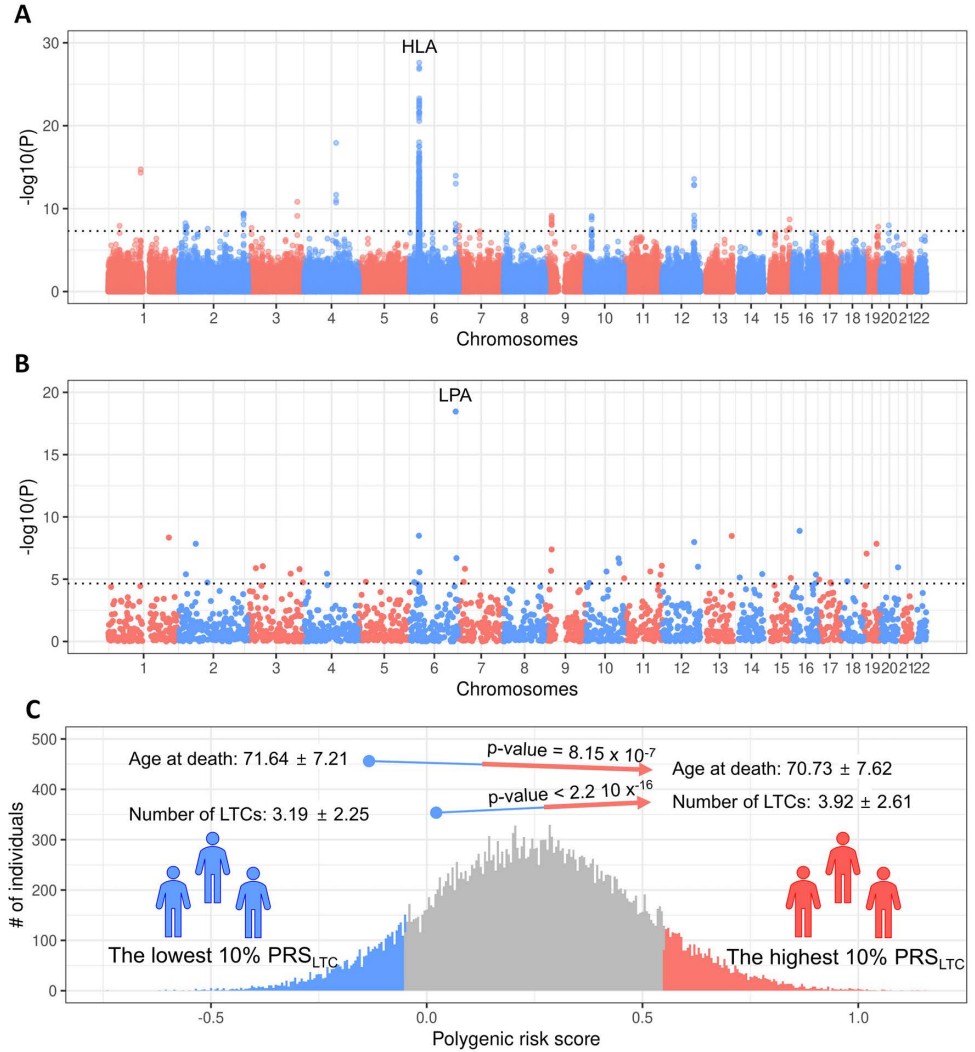

**Fig 1. Genetics of LTC burden and its association with lifespan.** Panel A shows a Manhattan plot of GWAS results for LTC burden in the training set (n = 343,868). Each point represents a genetic variant plotted by chromosomal position (x-axis) and $-\log_{10}$ p-value (y-axis). The dashed horizontal line marks the genome-wide significance threshold ($p = 5 \times 10^{-8}$). Panel B shows a Manhattan plot of local genetic correlations between LTC burden and parental lifespan, with each point representing a genomic region. The dashed line marks the Bonferroni-corrected significance threshold ($p = 2.23 \times 10^{-5}$). The most significant regions include the HLA (Panel A) and *LPA* (Panel B) loci. Panel C shows the distribution of the polygenic risk score for LTC burden (PRS_LTC) in the validation set (n = 34,339). The lowest 10% (blue) and highest 10% (red) PRS_LTC groups are compared for age at death and number of LTCs, with mean ± SD values and p-values shown for group differences.

To examine the association between genetic risk for the LTC burden and lifespan, we derived a PRS$_{LTC}$ in the validation set based on the GWAS result for LTC burden from the training set. The PRS$_{LTC}$ was positively correlated with LTC burden (Spearman's coefficient = 0.081, p-value < $2.2 \times 10^{-16}$) and negatively correlated with age at death (Spearman's coefficient = −0.032, p-value = $2.53 \times 10^{-9}$). Individuals in the highest 10% of the PRS$_{LTC}$ had, on average, 0.73 more LTCs at the time of recruitment (p-value < $2.2 \times 10^{-16}$) and a 0.9-year shorter lifespan (p-value = $8.15 \times 10^{-7}$) compared to individuals with the lowest 10% of the PRS$_{LTC}$ (Fig 1C). Although statistically significant, these differences were modest in magnitude.

In sex-stratified analyses, males in the highest $PRS_{LTC}$ had a shorter average lifespan (70.80 years, SD = 7.53) compared with those in the lowest decile (71.92 years, SD = 7.04; p-value = 2.13 x 10^{-6}), while females showed a similar but weaker and not significant difference (70.62 years, SD = 7.77 vs. 71.28 years, SD = 7.46; p-value = 0.041). The LTC burden increased with higher $PRS_{LTC}$ in both sexes, rising from 3.31 (SD = 2.29) to 4.09 (SD = 2.62) in males (p-value < 2.2 x 10^{-16}) and from 3.01 (SD = 2.18) to 3.67 (SD = 2.54) in females (p-value = 1.10 x 10^{-11}).

## Discussion

The most significant loci identified in the GWAS conducted in this study were located in the HLA region, which is known to play an important role in immunity and inflammation and has been linked to various long-term conditions [25–28]. Given the previously reported associations between immunity, inflammation, and MLTC, our findings provide genetic-level insights into the relationship between immune function and LTC burden [9–11,29]. The local genetic correlation analysis also revealed the most significant signal in the LPA gene region. The LPA gene has been reported to have genetic associations with coronary artery disease, diabetes, and depression, and has also been shown to have both phenotypic and genetic links to lifespan [7,30–33]. The local genetic correlation observed in the LPA gene region between LTC burden and lifespan in this study may offer genetic insights into the lethal, lifespan-shortening components of MLTC.

The associations observed at the *HLA* and *LPA* loci may indicate that both inflammation and lipid metabolism contribute to ageing and LTC burden. Variants in *HLA* can influence immune activation and chronic inflammation, while *LPA* variants regulate lipoprotein(a) levels, a known risk factor for atherosclerosis and vascular disease [7]. Previous studies suggest that these mechanisms may act through independent pathways but could also interact through shared mechanisms such as endothelial dysfunction and oxidative stress [7,34,35]. Although our study was not designed to test this directly, these findings highlight the potential link between immune and lipid pathways in the development of age-related multimorbidity. Future studies should investigate whether *HLA* and *LPA* act synergistically or independently to influence disease progression, aging, and lifespan.

The observed associations between $PRS_{LTC}$, LTC burden, and lifespan suggest a shared genetic architecture between the accumulation of LTCs and reduced longevity. However, the magnitude of these associations was modest. This may indicate that while genetic factors contribute to both LTC burden and lifespan, environmental and lifestyle factors are also likely to play a substantial role.

This study provides insights into the genetics of LTC burden and its association with lifespan using a large-scale dataset with systematic ascertainment of long-term conditions. However, several limitations warrant cautious interpretation. First, there are various ways to define each LTC, and this study only included a subset of common LTCs based on hospital inpatient records. This reliance on hospital-derived phenotypes may introduce misclassification, incomplete case capture, and censoring, potentially affecting the accuracy of the LTC burden estimates. Second, the interplay between different LTCs, lifestyle, and environmental factors—each of which may amplify or mitigate disease impact relative to a simple summation of conditions—was not examined in the present study. Third, analyses were limited to individuals of European ancestry, restricting generalisability and underscoring the need for replication in diverse populations. Fourth, the genetic correlation analyses should be interpreted as reflecting shared genetic signal rather than causality, and their accuracy may be affected by factors such as LD reference panels, block definitions, SNP density, and local heritability estimates. Fifth, although comprehensive genotype QC and imputation procedures were applied, inaccuracies may persist—particularly in regions with low SNP density—which could influence downstream analyses. In addition, while the PRS analyses provided meaningful associations, the predictive performance of PRS for complex traits such as LTC burden is generally modest and should be interpreted accordingly. Finally, LTC burden as defined here aggregates heterogeneous conditions, ranging from relatively mild and common to severe and life-limiting, meaning that the observed genetic signals may reflect shared biological pathways as well as strong effects attributable to specific high-impact LTCs. Future studies should evaluate stratified or severity-weighted LTC burdens, incorporate

environmental and sociodemographic factors, and use multi-ethnic datasets to more comprehensively characterise the genetic and non-genetic determinants of multimorbidity.

## Supporting information

**S1 Table. Baseline characteristics of the study population.**
(XLSX)

**S2 Table. Independent genome-wide significant loci from the GWAS on LTC burden.**
(XLSX)

**S3 Table. Annotations of associated variants.**
(XLSX)

**S4 Table. Local genetic correlations between the LTC burdens and parental lifespan.**
(XLSX)

## Acknowledgments

This work used the Edinburgh Compute and Data Facility (ECDF) (http://www.ecdf.ed.ac.uk/). This research has been conducted using the UK Biobank Resource under Application Number 44986. This work uses data provided by patients and collected by the NHS as part of their care and support. For open access, the author has applied a CC-BY public copyright licence to any Author Accepted Manuscript version arising from this submission.

## Author contributions

**Conceptualization:** Youngjune Bhak, Bruce Guthrie, Albert Tenesa.

**Data curation:** Youngjune Bhak.

**Formal analysis:** Youngjune Bhak.

**Funding acquisition:** Bruce Guthrie, Albert Tenesa.

**Investigation:** Youngjune Bhak.

**Methodology:** Youngjune Bhak.

**Project administration:** Youngjune Bhak, Albert Tenesa.

**Resources:** Youngjune Bhak.

**Software:** Youngjune Bhak.

**Supervision:** Bruce Guthrie, Albert Tenesa.

**Visualization:** Youngjune Bhak.

**Writing – original draft:** Youngjune Bhak.

**Writing – review & editing:** Youngjune Bhak, Bruce Guthrie, Albert Tenesa.

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
