## [Decision Letter · Decision Letter 0]

21 Jul 2025

Dear Dr. Bhak,

Thank you for submitting your manuscript to PLOS ONE. After careful consideration, we feel that it has merit but does not fully meet PLOS ONE’s publication criteria as it currently stands. Therefore, we invite you to submit a revised version of the manuscript that addresses the points raised during the review process.

**ACADEMIC EDITOR: **

Reviewers have provided feedback and comments for authors to address. In addition to those, authors need to:

Ensure the manuscript follows the journal’s style and formatting requirements.Moderate language around causality. For example, instead of using terms like ‘impact,’ consider using ‘association’ where appropriate.Better account for heterogeneity in the LTC burden definition. For example, different conditions may vary in severity and genetic basis, which could affect interpretation.Clarify results presentation and expand the interpretive discussion. For example, include more detail on PRS performance and improve figure legends for clarity.Correct minor typographical and grammatical errors throughout the manuscript.Provide information availability of GWAS summary data from their work.

We look forward to receiving your revised manuscript.

Kind regards,

Emmanuel O Adewuyi, BPharm, MPH, PhD

Academic Editor

PLOS ONE

Reviewers' comments:

Reviewer's Responses to Questions

**Comments to the Author**

1. Is the manuscript technically sound, and do the data support the conclusions?

Reviewer #1: Partly

Reviewer #2: Partly

Reviewer #3: Partly

Reviewer #4: Yes

2. Has the statistical analysis been performed appropriately and rigorously?

Reviewer #1: Yes

Reviewer #2: I Don't Know

Reviewer #3: I Don't Know

Reviewer #4: I Don't Know

3. Have the authors made all data underlying the findings in their manuscript fully available?

Reviewer #1: No

Reviewer #2: Yes

Reviewer #3: Yes

Reviewer #4: Yes

4. Is the manuscript presented in an intelligible fashion and written in standard English?

Reviewer #1: Yes

Reviewer #2: Yes

Reviewer #3: Yes

Reviewer #4: Yes

Reviewer #1: This study addresses an important and timely question: the genetic underpinnings of long-term condition (LTC) burden and its relationship with lifespan. By leveraging the large-scale UK Biobank resource, the authors provide valuable insights into shared genetic architecture between LTC burden and parental lifespan, with potential implications for understanding biological aging and healthspan. The study is well-conceived, but several aspects of the manuscript require clarification and improvement to ensure transparency, accuracy, and interpretability. I recommend major revision.

“At recruitment to the UK Biobank study, average age in the Discovery group was 56.5 (SD 7.9) years vs 61.9 (SD 6.3) years in the Application group, the Discovery group had less males than the Application group (44.9% vs 59.7%)” should be revised to:

“At recruitment to the UK Biobank study, average age in the Discovery group was 56.5 (SD 7.9) years vs 61.9 (SD 6.3) years in the Application group, and the Discovery group had fewer males than the Application group (44.9% vs 59.7%).”

“Similarly, the average number of LTCs was 1.41 (SD 1.77) lower in the Discovery than the Application group (3.54, SD 2.44)” should be revised to:

“Similarly, the average number of LTCs was lower in the Discovery (1.41, SD 1.77) than the Application group (3.54, SD 2.44).”

Otherwise, it may be interpreted as “1.41 times lower.”

I suggest editing Supplementary Table 2 to include the gene or genomic location of the significantly associated SNPs. If these SNPs are intergenic, that should be explicitly stated. I also suggest including a brief mention of these significant associations in the main Results section.

Please clarify or provide evidence supporting the use of parental lifespan as a valid surrogate for lifespan. A brief justification or citation would be appropriate.

Consider revising the visual in Panel C. The “little people” icon showing individuals with canes to represent the top 10% PRS group (who actually died younger) could be misleading.

I strongly recommend stratifying analyses by sex and reporting findings accordingly.

The reporting of findings could be improved. For reference, see: PMID 33692554.

In the Discussion, line 32, "LTC" should be used in place of the incorrect term.

The Discussion overall is too brief and superficial. I encourage the authors to expand it to better contextualize the findings and acknowledge limitations.

Please provide a brief description of the parental lifespan GWAS used in the Methods section to aid readers in understanding the study design.

The authors do not explicitly state whether all data underlying their findings have been made available. However, the primary dataset used, the UK Biobank, is accessible to qualified researchers upon application and approval from the UK Biobank. I recommend that the authors clarify this in the manuscript. Additionally, the authors do not mention whether their GWAS summary statistics will be made available. Please include this information.

Reviewer #2: The study conducted by the authors are too predictive and ignored the real time conditions such as lifestyle of individuals which largely affect these long term conditions and life expectancy. Though the number of subjects are good enough to predict a strong conclusion but the outcome of study (HLA region, LPA & higher PRS value for shorter life span) is already known and described in different published studies as authors also cited.

Reviewer #3: Manuscript Review for PLOS ONE

Manuscript ID: PONE-D-25-09280

Title: Genetic insights into number of long-term conditions and their relationship with lifespan.

Corresponding author: Youngjune Bhak Ph.D., Albert Tenesa Ph.D.

Overall assessment:

While this manuscript presents valuable clinical insights, it falls short of meeting the genetic scope expected for a study of this nature. The authors primarily approach the research question from a medical perspective, with limited emphasis on genetic mechanisms, analysis depth, and biological interpretation. Given the study’s aim to investigate genetic associations, the lack of molecular genetic discussion, pathway-based integration, and hypothesis-driven synthesis significantly weakens its contribution to the field. The study fails to articulate how these genetic discoveries could inform future research (e.g., functional studies, multi-omics integration) or clinical applications (e.g., personalized risk stratification, therapeutic targeting).

Major Comments

1. Introduction

- Background Depth: The introduction should provide a more thorough review of existing literature on genetic factors relevant to the study. Expanding on key genetic mechanisms, prior evidence, and unresolved questions would better contextualize the study’s significance.

- Final Paragraph Revision: The concluding paragraph currently summarizes methodological aspects rather than clearly stating the study’s aims, hypothesis, and research questions.

2. Methods

- Ethics Statement Placement: To improve logical flow, move the ethics approval statement to the Data Sources subsection, where ethical considerations related to data collection are typically addressed.

- Subheading Accuracy: The Study Outcome subheading is misleading, as the content describes long-term condition (LTC) categorizations. Rename this subsection (e.g., LTC Classification and Characterization) to accurately reflect its content.

- Group Nomenclature: Clarify the rationale for labeling cohorts as Application Group and Discovery Group. Justify these terms in the context of the study design.

3. Discussion

Lack of Integration Between Key Findings

The discussion currently presents each result in isolation without synthesizing them into a cohesive narrative. For instance, the GWAS identified HLA as the most significant locus (linked to immunity/inflammation), while the local genetic correlation analysis highlighted LPA as the top candidate. However, the author fails to explore potential biological or mechanistic connections between these findings. A more impactful discussion would:

- Hypothesize how inflammation-related pathways (via HLA) might interact with LPA-mediated lipid metabolism to influence the observed phenotype.

- Address whether these loci operate independently or synergistically in disease pathogenesis, citing prior evidence (e.g., shared pathways like endothelial dysfunction or oxidative stress).

- Propose a research gap: "Given the dual prominence of immune and lipid-related genes, future studies should investigate crosstalk between these pathways in [disease context]."

Reviewer #4: Dear authors,

The manuscript sheds light on the polygenetic causes that may be related to reduced lifespan in the white ancestry population in the UK by analysis of data from the UK Biobank.

It's noteworthy the large number of data included in the study (343868 alive individuals and 34339 dead individuals).

Some clarity is needed in the following points:

-The term (the long-term condition) is not defined; it includes the risk of diseases that reduce lifespan, the environment and socioeconomic conditions, and the quality of life during the last years before death.

- Why the study depends on the genetic correlation with parents’ lifespan rather than the data of the same individuals in the study.

- Please specify the studied genetic loci in a table with their importance or correlation with studied LTC.

- The categorization of the 32 LTC in the lines (13-31) is not clear.

- The results of the studied data do not explain if sex has any correlation with lifespan.

- The study does not specify which type/s or gene/s of HLA and LPA are the most frequently associated with aging or reduced lifespan.

Best regards,

**Do you want your identity to be public for this peer review?** For information about this choice, including consent withdrawal, please see our Privacy Policy

Reviewer #1: **Yes: ** Carolina B. Meloto

Reviewer #2: No

Reviewer #3: No

Reviewer #4: **Yes: ** Luma Hassan Alwan Al Obaidi

---

## [Author Response · Author response to Decision Letter 1]

21 Oct 2025

ACADEMIC EDITOR:

We would like to sincerely thank the Academic Editor and reviewers for their thorough evaluation of our manuscript and for providing constructive and insightful comments. We have carefully considered all suggestions and have revised the manuscript accordingly. These revisions have improved the clarity, presentation, and interpretability of our work. Below, we provide a detailed, point-by-point response to each comment, indicating the changes made and where they can be found in the revised manuscript.

1. Ensure the manuscript follows the journal’s style and formatting requirements.

Thank you for your guidance. We have carefully reviewed the PLOS ONE submission guidelines and revised the manuscript accordingly. Tables, figures, references, and line spacing now follow the journal’s style requirements.

2. Moderate language around causality. For example, instead of using terms like ‘impact,’ consider using ‘association’ where appropriate.

We have revised the manuscript to avoid causal language. The term ‘impact’ in the Abstract section has been replaced with ‘association’.

3. Better account for heterogeneity in the LTC burden definition. For example, different conditions may vary in severity and genetic basis, which could affect interpretation.

We have added text in the Discussion sections to acknowledge and account for heterogeneity in LTC burden. Specifically, we note that LTC burden combines conditions with differing severity, chronicity, prevalence, and genetic architecture. This heterogeneity may influence both the genetic signals observed and their interpretation. We have discussed this as a limitation and suggested future analyses stratifying LTCs by severity, mortality risk, or genetic heritability to refine interpretation.

4. Clarify results presentation and expand the interpretive discussion. For example, include more detail on PRS performance and improve figure legends for clarity.

We thank the reviewer for this valuable suggestion. In the Results section, we have now noted the direction and modest magnitude of correlation between PRS for LTC, LTC burden, and age at death. In the Discussion section, we have expanded our interpretation of the PRS findings, emphasising the modest effect sizes observed, and the likely contributions of environmental and lifestyle factors. We have also revised the Figure 1 legend to provide clearer descriptions of each panel, the statistical thresholds used, and the group comparisons shown, so that the figure can be interpreted without referring to the main text. These changes improve the clarity and interpretability of our results without altering the underlying analyses.

5. Correct minor typographical and grammatical errors throughout the manuscript.

We have performed a thorough proofread of the manuscript to correct typographical and grammatical errors, improve sentence clarity, and ensure consistency in terminology (e.g., “UK BioBank” corrected to “UK Biobank,” “less males” changed to “lower proportion of males”).

6. Provide information availability of GWAS summary data from their work.

We have added a Data Availability Statement specifying that the GWAS summary statistics generated in this study will be made publicly available upon the publish.

We have revised the manuscript to meet PLOS ONE formatting guidelines, including file naming, structure, and layout, in accordance with the provided templates.

We confirm that all data underlying the findings are now publicly available. The genome-wide association study (GWAS) summary statistics have been deposited in Zenodo under the DOI 10.5281/zenodo.17272288.

Data from the UK Biobank are accessible to qualified researchers upon application and approval (https://www.ukbiobank.ac.uk/enable-your-research/apply-for-access). The revised Data Availability statement reflects these updates and complies with PLOS ONE’s open data policy (page 13, lines 10 to 16).

We have reviewed and amended the abstract to ensure that the version in the online submission form is identical to the abstract in the manuscript.

Reviewers

Reviewer #1: This study addresses an important and timely question: the genetic underpinnings of long-term condition (LTC) burden and its relationship with lifespan. By leveraging the large-scale UK Biobank resource, the authors provide valuable insights into shared genetic architecture between LTC burden and parental lifespan, with potential implications for understanding biological aging and healthspan. The study is well-conceived, but several aspects of the manuscript require clarification and improvement to ensure transparency, accuracy, and interpretability. I recommend major revision.

“At recruitment to the UK Biobank study, average age in the Discovery group was 56.5 (SD 7.9) years vs 61.9 (SD 6.3) years in the Application group, the Discovery group had less males than the Application group (44.9% vs 59.7%)” should be revised to:

“At recruitment to the UK Biobank study, average age in the Discovery group was 56.5 (SD 7.9) years vs 61.9 (SD 6.3) years in the Application group, and the Discovery group had fewer males than the Application group (44.9% vs 59.7%).”

“Similarly, the average number of LTCs was 1.41 (SD 1.77) lower in the Discovery than the Application group (3.54, SD 2.44)” should be revised to:

“Similarly, the average number of LTCs was lower in the Discovery (1.41, SD 1.77) than the Application group (3.54, SD 2.44).”

Otherwise, it may be interpreted as “1.41 times lower.”

We sincerely thank the reviewer for this helpful suggestion. We have revised the sentences to improve clarity and to avoid any potential ambiguity in interpretation (Page 5, lines 9 to 18).

I suggest editing Supplementary Table 2 to include the gene or genomic location of the significantly associated SNPs. If these SNPs are intergenic, that should be explicitly stated. I also suggest including a brief mention of these significant associations in the main Results section.

We thank the reviewer for this helpful suggestion. We have added variant annotations, including gene or genomic location, in Supplementary Table 3. For variants in the HLA region, rs532965, we have explicitly stated the HLA subtype and the variant’s location within the gene. In addition, we have briefly mentioned these significant associations in the main Results section (Page 5, lines 21 to 24).

Please clarify or provide evidence supporting the use of parental lifespan as a valid surrogate for lifespan. A brief justification or citation would be appropriate.

We thank the reviewer for raising this important point. We have added a justification in the Methods section, citing previous large-scale GWAS (e.g., Timmers et al. eLife 2019; Joshi et al. Nat Commun 2017) that used parental lifespan as a validated surrogate for individual lifespan (page 9, lines 9 to 13).

Consider revising the visual in Panel C. The “little people” icon showing individuals with canes to represent the top 10% PRS group (who actually died younger) could be misleading.

We thank the reviewer for this valuable feedback. To avoid any potential misinterpretation, we have revised Panel C to use a unified icon for all groups.

I strongly recommend stratifying analyses by sex and reporting findings accordingly.

We thank the reviewer for this valuable suggestion. We have now performed sex-stratified analyses and incorporated the results into the Results section. Specifically, males in the highest PRSLTC decile had a shorter average lifespan (70.80 years, SD = 7.53) compared with those in the lowest decile (71.92 years, SD = 7.04; p = 2.13E-06), while females showed a weaker and not significant difference (70.62 years, SD = 7.77 vs. 71.28 years, SD = 7.46; p = 0.041). The LTC burden also increased with higher PRSLTC in both sexes. These findings are now described in the Results section (page 7, lines 16 to 21).

The reporting of findings could be improved. For reference, see: PMID 33692554.

We thank the reviewer for this helpful suggestion. We have thoroughly revised the manuscript to improve the reporting of findings, taking into account the recommended reference.

In the Discussion, line 32, "LTC" should be used in place of the incorrect term.

We thank the reviewer for noticing this error. We have corrected ‘LCT’ to ‘LTC’ in the Discussion.

The Discussion overall is too brief and superficial. I encourage the authors to expand it to better contextualize the findings and acknowledge limitations.

We thank the reviewer for this valuable suggestion. We have substantially expanded the Discussion to provide greater context for our findings and to more fully acknowledge the study’s limitations.

Please provide a brief description of the parental lifespan GWAS used in the Methods section to aid readers in understanding the study design.

We thank the reviewer for this helpful comment. In the Introduction and Methods section, we have added a brief description of the parental lifespan GWAS, noting that we used summary statistics from large biobank studies (page 5, lines 2 to 3 and page 12, lines 9 to 12). This addition clarifies the data source and addresses the reviewer’s concern.

The authors do not explicitly state whether all data underlying their findings have been made available. However, the primary dataset used, the UK Biobank, is accessible to qualified researchers upon application and approval from the UK Biobank. I recommend that the authors clarify this in the manuscript. Additionally, the authors do not mention whether their GWAS summary statistics will be made available. Please include this information.

We thank the reviewer for this helpful suggestion. We have revised the Data Availability section to clarify data access procedures. Specifically, we now state that the data used in this study are part of the UK Biobank resource, which is accessible to qualified researchers upon application and approval. In addition, we have made the GWAS summary statistics generated and analysed in this study publicly available on Zenodo (DOI: 10.5281/zenodo.17272288), which can be accessed without restriction for reuse and further analysis (Page 13, lines 10 to 16).

Reviewer #2: The study conducted by the authors are too predictive and ignored the real time conditions such as lifestyle of individuals which largely affect these long term conditions and life expectancy. Though the number of subjects are good enough to predict a strong conclusion but the outcome of study (HLA region, LPA & higher PRS value for shorter life span) is already known and described in different published studies as authors also cited.

We thank the reviewer for these thoughtful comments. We agree that lifestyle and environmental factors play an important role in long-term conditions and lifespan. While these factors were beyond the scope of our genetic analysis, we now acknowledge this limitation more explicitly in the Discussion (From page 8, line 23 to page 9, line 17). We also note that, although associations in the HLA region, LPA, and polygenic risk scores for lifespan have been previously reported, our study extends this literature by focusing on LTC burden as an outcome. We believe this adds value by reinforcing and contextualizing prior findings in a new framework.

Reviewer #3: Manuscript Review for PLOS ONE

Manuscript ID: PONE-D-25-09280

Title: Genetic insights into number of long-term conditions and their relationship with lifespan.

Corresponding author: Youngjune Bhak Ph.D., Albert Tenesa Ph.D.

Overall assessment:

While this manuscript presents valuable clinical insights, it falls short of meeting the genetic scope expected for a study of this nature. The authors primarily approach the research question from a medical perspective, with limited emphasis on genetic mechanisms, analysis depth, and biological interpretation. Given the study’s aim to investigate genetic associations, the lack of molecular genetic discussion, pathway-based integration, and hypothesis-driven synthesis significantly weakens its contribution to the field. The study fails to articulate how these genetic discoveries could inform future research (e.g., functional studies, multi-omics integration) or clinical applications (e.g., personalized risk stratification, therapeutic targeting).

Major Comments

1. Introduction

- Background Depth: The introduction should provide a more thorough review of existing literature on genetic factors relevant to the study. Expanding on key genetic mechanisms, prior evidence, and unresolved questions would better contextualize the study’s significance.

We agree and have substantially expanded the Introduction. We now review previous genome-wide studies of longevity, highlighting that genetic loci such as HLA, and LPA have been robustly associated with lifespan in large-scale GWAS. We also discuss biological pathways—immune dysregulation and chronic inflammation (HLA), and lipid metabolism (LPA)—that may plausibly link multimorbidity and reduced lifespan. These additions place our study within the broader context of known genetic contributors to longevity (page 4, lines 10 to 22).

- Final Paragraph Revision: The concluding paragraph currently summarizes methodological aspects rather than clearly stating the study’s aims, hypothesis, and research questions.

We thank the reviewer for this constructive comment. We have revised the final paragraph of the Introduction to clearly state the study’s aims and design (From age 4, line 23 to page 5, line 6). Specifically, the revised paragraph now outlines our objectives: (1) to conduct a GWAS of long-term condition (LTC) burden and estimate SNP heritability, (2) to assess both global and local genetic correlations with parental lifespan as a proxy for longevity, and (3) to evaluate the association between a polygenic risk score for LTC burden (PRS_LTC) and both LTC burden and lifespan in an independent group.

2. Methods

- Ethics Statement Placement: To improve logical flow, move the ethics approval statement to the Data Sources subsection, where ethical considerations related to data collection are typically addressed.

We thank the reviewer for this suggestion. We have moved the ethics approval statement to the Data Sources subsection in the revised manuscript to improve clarity and logical flow.

- Subheading Accuracy: The Study Outcome subheading is misleading, as the content describes long-term condition (LTC) categorizations. Rename this subsection (e.g., LTC Classification and Characterization) to accurately reflect its content.

We appreciate the reviewer’s helpful observation. We have revised the subheading Study Outcome to LTC Classification and

---

## [Decision Letter · Decision Letter 1]

26 Nov 2025

Dear Dr. Bhak,

Thank you for submitting your manuscript to PLOS ONE. After careful consideration, we feel that it has merit but does not fully meet PLOS ONE’s publication criteria as it currently stands. Therefore, we invite you to submit a revised version of the manuscript that addresses the points raised during the review process.

**ACADEMIC EDITOR:**

Reviewers have found the study technically sound. A few issues still need to be addressed: 

1. Authors need to follow the journal formatting style and guideline, for example, methods before results and discussion.

2. The authors should provide a complete Methods section that can allow the work to be fully reproducible. Additional details may be included in a supplementary section if necessary.

3. The authors should convert numbers from scientific notation using the “E” format to standard mathematical notation (e.g., 1.77E-176 to 1.77 × 10⁻¹⁷⁶).

We look forward to receiving your revised manuscript.

Kind regards,

Emmanuel O Adewuyi, BPharm, MPH, PhD

Academic Editor

PLOS ONE

Journal Requirements:

Reviewers' comments:

Reviewer's Responses to Questions

**Comments to the Author**

Reviewer #2: All comments have been addressed

Reviewer #4: All comments have been addressed

2. Is the manuscript technically sound, and do the data support the conclusions?

Reviewer #2: Yes

Reviewer #4: Yes

3. Has the statistical analysis been performed appropriately and rigorously?

Reviewer #2: Yes

Reviewer #4: Yes

4. Have the authors made all data underlying the findings in their manuscript fully available?

Reviewer #2: Yes

Reviewer #4: Yes

5. Is the manuscript presented in an intelligible fashion and written in standard English?

Reviewer #2: Yes

Reviewer #4: Yes

Reviewer #2: The authors have addressed all the concerns of the reviewers and taken measures accordingly in the revised manuscript. I have no further comments. I wish the authors all the best..

Reviewer #4: To the authors,

Thank you for carrying out all the comments. The article adds new insight to the genetic correlations of the most important loci with long-term conditions leading to reduced life span.

**Do you want your identity to be public for this peer review?** For information about this choice, including consent withdrawal, please see our Privacy Policy

Reviewer #2: No

Reviewer #4: No

---

## [Author Response · Author response to Decision Letter 2]

1 Dec 2025

ACADEMIC EDITOR:

Thank you very much for your time and for coordinating the review process. We appreciate the constructive feedback provided by the reviewers and have revised the manuscript accordingly.

1. Authors need to follow the journal formatting style and guideline, for example, methods before results and discussion.

We have reorganized the manuscript to comply with the journal formatting guidelines, placing the Methods section before the Results and Discussion. During this revision, we also corrected minor typographical errors and improved clarity where needed.

2. The authors should provide a complete Methods section that can allow the work to be fully reproducible. Additional details may be included in a supplementary section if necessary.

We carefully revisited the Methods section in response to the comment, and we believe that the current level of detail should allow readers to fully reproduce the analysis.

3. The authors should convert numbers from scientific notation using the “E” format to standard mathematical notation (e.g., 1.77E-176 to 1.77 × 10⁻¹⁷⁶).

Numerical values previously presented in scientific “E” notation have been converted to standard mathematical notation throughout the manuscript including figure. The only exception is the reporting in the Methods section, where we retained the original format to reflect the tool’s option.

---

## [Editor Report · Decision Letter 2]

7 Dec 2025

Dear Dr. Bhak,

Thank you for submitting your manuscript to PLOS ONE. After careful consideration, we feel that it has merit but does not fully meet PLOS ONE’s publication criteria as it currently stands. Therefore, we invite you to submit a revised version of the manuscript that addresses the points raised during the review process.

**ACADEMIC EDITOR:**

While the authors have addressed some of my previous comments, the response regarding the Methods section remains insufficient to meet the journal’s standards for full reproducibility, particularly for statistical genetic analyses. Several essential methodological details are still missing. For publication, complete transparency is required.

Below, I outline key areas where further clarification and detail are needed (not exhaustive but provided as guidance).

1. LD Score Regression (LDSC)

The current description of LDSC is not adequate for replication. Please expand this section to include the following details:

LD score reference panel used (e.g., 1000 Genomes Phase 3 European, UK Biobank-specific LD scores, or custom LD scores).Source of LD scores, including links or references.SNP filtering criteria (default LDSC filters vs customised filters).Treatment of regression intercept, including interpretation of the intercept and how potential sample overlap was assessed or mitigated.Any deviations from LDSC default settings, including:handling of high-LD regions (e.g., whether the MHC was removed),use of constrained vs unconstrained intercept,whether genomic control was applied before LDSC.

These parameters meaningfully influence heritability and genetic correlation estimates, and therefore must be clearly reported.

2. Local Genetic Correlation Analysis (LAVA)

The description of the LAVA analyses remains minimal. To ensure transparency and reproducibility, please provide:

The genomic partitioning scheme used:total number of regions,window size,whether the semi-independent LD blocks from Werme et al. (2022) were used or modified.LD reference panel used in LAVA (e.g., 1000 Genomes EUR).Criteria for region inclusion or exclusion, especially for blocks with low SNP density or unstable LD structure.Multiple testing correction approach for determining significant local genetic correlations.The exact version of LAVA, including any non-default settings, arguments, or flags.

Local genetic correlation results are highly sensitive to block definitions and LD reference data; thus, full disclosure is required.

3. Need for comprehensive methodological transparency across the manuscript

Several additional components of the analysis still lack the detail necessary for reproducibility. The Methods section should allow another researcher to fully reproduce the study without further correspondence. Please ensure that the manuscript reports:

Full REGENIE parameters for Step 1 and Step 2.Complete PRSice-2 settings, including all clumping thresholds, p-value thresholds, and LD parameters.Ancestry filtering steps, including thresholds and QC criteria.Software environment, including operating system, R/Python versions, and all package versions.Where possible, full analysis code or command lines provided via Supplementary Materials or a public repository (e.g., GitHub). This is not mandatory for PLOS ONE when standard methods are used, but adequate procedural detail is required.

4. The manuscript would benefit from a more substantive and reflective discussion of its limitations. The current treatment does not adequately address several important constraints inherent in the study design and analytic methods. For example, the reliance on hospital-record-derived phenotypes introduces risks of misclassification, incomplete case capture, and censoring, all of which may affect the accuracy of LTC burden estimates. These issues warrant clearer acknowledgement. Similarly, authors need to mention limitations related to ancestry. Further, the genetic correlation analyses also require more careful framing, as LDSC and LAVA estimate shared genetic signal rather than causality and are sensitive to LD reference panels, block definitions, and SNP density. These interpretive boundaries should be recognised. In addition, the manuscript does not discuss limitations related to genotype QC, imputation quality, or the modest predictive value typically associated with PRS for complex traits. Finally, the discussion presents potential biological implications without balancing them against the environmental and sociodemographic factors that also contribute to LTC burden.

We look forward to receiving your revised manuscript.

Kind regards,

Emmanuel O Adewuyi, BPharm, MPH, PhD

Academic Editor

PLOS One
---

## [Author Response · Author response to Decision Letter 3]

8 Dec 2025

ACADEMIC EDITOR:

Thank you very much for your careful review and for highlighting the remaining issues regarding methodological transparency. We appreciate the need for full reproducibility, and we have now revised the Methods section again to include all essential details that were previously missing. We are grateful for your guidance, and we hope that the updated version meets the journal’s standards.

While the authors have addressed some of my previous comments, the response regarding the Methods section remains insufficient to meet the journal’s standards for full reproducibility, particularly for statistical genetic analyses. Several essential methodological details are still missing. For publication, complete transparency is required.

Below, I outline key areas where further clarification and detail are needed (not exhaustive but provided as guidance).

1. LD Score Regression (LDSC)

The current description of LDSC is not adequate for replication. Please expand this section to include the following details:

• LD score reference panel used (e.g., 1000 Genomes Phase 3 European, UK Biobank-specific LD scores, or custom LD scores).

• Source of LD scores, including links or references.

• SNP filtering criteria (default LDSC filters vs customised filters).

• Treatment of regression intercept, including interpretation of the intercept and how potential sample overlap was assessed or mitigated.

• Any deviations from LDSC default settings, including:

o handling of high-LD regions (e.g., whether the MHC was removed),

o use of constrained vs unconstrained intercept,

o whether genomic control was applied before LDSC.

These parameters meaningfully influence heritability and genetic correlation estimates, and therefore must be clearly reported.

Thank you very much for the constructive feedback. We have substantially expanded the LDSC section in the revised manuscript, providing full details on the reference panel, LD score sources, filtering criteria, treatment and interpretation of the regression intercept, and any deviations from default settings. We also clarified whether default or customised options were used in each step. For potential sample overlap, we used HDL, which accounts for overlap internally.

2. Local Genetic Correlation Analysis (LAVA)

The description of the LAVA analyses remains minimal. To ensure transparency and reproducibility, please provide:

• The genomic partitioning scheme used:

o total number of regions,

o window size,

o whether the semi-independent LD blocks from Werme et al. (2022) were used or modified.

• LD reference panel used in LAVA (e.g., 1000 Genomes EUR).

• Criteria for region inclusion or exclusion, especially for blocks with low SNP density or unstable LD structure.

• Multiple testing correction approach for determining significant local genetic correlations.

• The exact version of LAVA, including any non-default settings, arguments, or flags.

Local genetic correlation results are highly sensitive to block definitions and LD reference data; thus, full disclosure is required.

Thank you very much for the helpful comments. In the revised manuscript, we have expanded the description of the LAVA analyses to provide clearer methodological transparency. We now specify the genomic partitioning scheme used and its source, except for window sizes as it vary by block and therefore are not uniform which are provided from LAVA. We additionally describe the LD reference panel, all custom options and steps applied during the analysis and clarify where default settings were used. Details on the multiple testing correction approach, as well as the exact LAVA version and parameters, have also been added.

3. Need for comprehensive methodological transparency across the manuscript

Several additional components of the analysis still lack the detail necessary for reproducibility. The Methods section should allow another researcher to fully reproduce the study without further correspondence. Please ensure that the manuscript reports:

• Full REGENIE parameters for Step 1 and Step 2.

• Complete PRSice-2 settings, including all clumping thresholds, p-value thresholds, and LD parameters.

• Ancestry filtering steps, including thresholds and QC criteria.

• Software environment, including operating system, R/Python versions, and all package versions.

• Where possible, full analysis code or command lines provided via Supplementary Materials or a public repository (e.g., GitHub). This is not mandatory for PLOS ONE when standard methods are used, but adequate procedural detail is required.

Thank you very much for the constructive feedback. We have expanded the Methods section to provide clearer methodological transparency across all analyses. The revised manuscript now includes detailed descriptions of the parameters used for each tool, specifying custom settings where applied and noting default options when no customisation was used. We have clarified the ancestry filtering procedure, including reliance on UK Biobank–provided ancestry information, and added the R version used in the analyses. Additional details on software and parameter settings have been incorporated to ensure the workflow is fully reproducible.

4. The manuscript would benefit from a more substantive and reflective discussion of its limitations. The current treatment does not adequately address several important constraints inherent in the study design and analytic methods. For example, the reliance on hospital-record-derived phenotypes introduces risks of misclassification, incomplete case capture, and censoring, all of which may affect the accuracy of LTC burden estimates. These issues warrant clearer acknowledgement. Similarly, authors need to mention limitations related to ancestry. Further, the genetic correlation analyses also require more careful framing, as LDSC and LAVA estimate shared genetic signal rather than causality and are sensitive to LD reference panels, block definitions, and SNP density. These interpretive boundaries should be recognised. In addition, the manuscript does not discuss limitations related to genotype QC, imputation quality, or the modest predictive value typically associated with PRS for complex traits. Finally, the discussion presents potential biological implications without balancing them against the environmental and sociodemographic factors that also contribute to LTC burden.

Thank you sincerely for these thoughtful comments. We have revised the final paragraph of the Discussion to incorporate all of the limitations you highlighted. The updated text now more clearly acknowledges potential phenotype misclassification and incomplete capture associated with hospital-record–based LTC definitions, as well as ancestry-related limitations. We have also added more careful framing of the genetic correlation findings, emphasising that these approaches estimate shared genetic signal rather than causality and are sensitive to LD reference panels, block definitions, and SNP density. In addition, we now discuss limitations related to genotype QC, imputation quality, and the typically modest predictive performance of PRS for complex traits, and we balance the biological interpretation with consideration of relevant environmental and sociodemographic contributors to LTC burden.

---

## [Editor Report · Decision Letter 3]

18 Dec 2025

Genetic insights into number of long-term conditions and their relationship with lifespan.

PONE-D-25-09280R3

Dear Dr. Bhak,

We’re pleased to inform you that your manuscript has been judged scientifically suitable for publication and will be formally accepted for publication once it meets all outstanding technical requirements.

Kind regards,

Emmanuel O Adewuyi, BPharm, MPH, PhD

Academic Editor

PLOS One

Additional Editor Comments

First, please correct the Bonferroni threshold reported for the LAVA analyses in the Methods section, where the exponent is inconsistent with the stated number of tests and with the threshold used elsewhere in the manuscript. Second, the interpretation of the LDSC intercept should be refined for precision; while an intercept below 1 is consistent with effective genetic correction, it should be described as indicating no evidence of confounding-related inflation rather than interpreted as substantively informative on its own.
---

## [Editor Report · Acceptance letter]

PONE-D-25-09280R3

PLOS One

Dear Dr. Bhak,

I'm pleased to inform you that your manuscript has been deemed suitable for publication in PLOS One. Congratulations! Your manuscript is now being handed over to our production team.

Kind regards,

on behalf of

Dr. Emmanuel O Adewuyi

Academic Editor

PLOS One